# Thermodynamic Study of the Solubility of Triclocarban in Polyethylene Glycol 200 + Water Cosolvent Mixtures at Different Temperatures

**DOI:** 10.3390/molecules30122631

**Published:** 2025-06-17

**Authors:** Vanesa Puentes-Lozada, Diego Ivan Caviedes-Rubio, Cristian Rincón-Guio, Nestor Enrique Cerquera, Rossember Edén Cardenas-Torres, Claudia Patricia Ortiz, Fleming Martinez, Daniel Ricardo Delgado

**Affiliations:** 1Programa de Maestría en Ingeniería y Gestión Ambiental, Universidad Surcolombiana, Avenida Pastrana Borrero, Carrera 1, Neiva 410001, Huila, Colombia; vanepl1204@hotmail.com; 2Grupo de Investigación de Ingenierías UCC-Neiva, Programa de Ingeniería Agroalimentaria, Facultad de Ingeniería, Universidad Cooperativa de Colombia, Sede Neiva, Calle 11 No. 1-51, Neiva 410001, Huila, Colombia; diego.caviedesr@campusucc.edu.co; 3Ingeniería Industrial, Institución Universitaria Politécnico Grancolombiano, Bogotá 110321, Cundinamarca, Colombia; calrincong@poligran.edu.co; 4Hidroingeniería y Desarrollo Agropecuario—GHIDA, Programa de Ingeniería Agrícola, Facultad de Ingeniería, Universidad Surcolombiana, Neiva 410001, Huila, Colombia; cerquera@usco.edu.co; 5Grupo de Energía Materiales y Diseño EnerDIMAT, Facultad de Ingeniería, Universidad de América, Av. Circunvalar No. 20-53, Bogotá 110321, Cundinamarca, Colombia; rossember.cardenas@uamerica.edu.co; 6Grupo de Investigación Ciencia, Ingeniería e Innovación, Carrera 8A No. 43-44, Neiva 410001, Huila, Colombia; cportizd14@gmail.com; 7Grupo de Investigaciones Farmacéutico-Fisicoquímicas, Departamento de Farmacia, Facultad de Ciencias, Universidad Nacional de Colombia, Sede Bogotá, Carrera 30 No. 45-03, Bogotá 110321, Cundinamarca, Colombia; 8Grupo de Investigación de Ingenierías UCC-Neiva, Programa de Ingeniería Civil, Facultad de Ingeniería, Universidad Cooperativa de Colombia, Sede Neiva, Calle 11 No. 1-51, Neiva 410001, Huila, Colombia

**Keywords:** solubility, triclocarban, cosolvency, solution thermodynamics, PEG 200

## Abstract

Background: Solubility is a fundamental physicochemical property in pharmaceutical, chemical and environmental industrial processes. Regarding Triclocarban (TCC), a broad-spectrum antimicrobial, solubility is particularly challenging due to its low aqueous solubility and hydrophobic nature; these challenges can be addressed by some effective techniques such as cosolvency, which allows one to increase the solubility of drugs by several orders of magnitude. This study aims to thermodynamically evaluate the solubility of TCC in cosolvent mixtures of PEG 200 + water at different temperatures. Methods: Experimental solubility data were determined using the shake-flask followed by UV quantification analysis at saturation methods, and thermodynamic functions of the solution processes were calculated using the Gibbs–van’t Hoff–Krug model. Results: The solubility results demonstrate the positive cosolvent effect of PEG 200 on the solubility of TCC, whose solution process is thermodynamically strongly governed by the enthalpy of solution with entropic preference in PEG 200-rich mixtures. Conclusions: The solubility of TCC is an endothermic, thermo-dependent process. The addition of PEG 200 to the cosolvent mixture favors this process and shows a positive cosolvent effect.

## 1. Introduction

Solubility is one of the most important physicochemical properties as it has profound implications both in pharmaceutical development and in the evaluation of the environmental performance of chemical compounds [1]. In the pharmaceutical field, the solubility of an active ingredient (API) is a critical determinant of its bioavailability [2]. Inadequate solubility can compromise therapeutic efficacy and require complex formulation strategies to ensure effective drug delivery [3]. In addition, solubility influences key processes such as purification, crystallization and overall dosage form design [4]. From an environmental perspective, the aqueous solubility of a compound determines its fate and transport in various environmental compartments [5]. It determines its partitioning between the aqueous and solid phases (soils, sediments) and influences its mobility, persistence and bioaccumulation potential [5]. Therefore, accurate solubility data are essential for modeling environmental fate, assessing ecosystem exposure and predicting the potential ecotoxicity of chemicals [6,7].

Triclocarban (TCC), (*N*-(4-chlorophenyl)-*N*’-(3,4-dichlorophenyl)urea, C_13_H_9_Cl_3_N_2_O) (Figure 1) is a synthetic antibacterial agent that has been widely used for decades, especially in bar and liquid soaps and other personal care products, due to its particular efficacy against Gram-positive bacteria [8,9,10]. However, the application and study of TCC faces a fundamental challenge: its extremely low water solubility, with reported values in the range of 0.045 to 11 mg/L (approximately 0.14 to 35 μM) at ambient temperatures, and its high hydrophobicity, reflected in an estimated octanol–water partition coefficient (Log *K*_*o*/*w*_) between 3.5 and 4.9 [11,12,13,14]. This low aqueous solubility not only hinders its formulation in aqueous systems, but also significantly affects its environmental performance [15,16,17]. In recent years, concerns have arisen about the persistence of TCC in the environment, with estimated half-lives of months to years in soils and sediments, and its potential as an endocrine disruptor [18,19]. These concerns have led to regulatory action, such as the U.S. FDA banning its use in certain OTC consumer antiseptic products in 2016 [20]. This context underscores the critical need for a thorough understanding of the physicochemical properties of TCC, in particular its solubility in relevant aqueous and mixed solvent systems [17].

Cosolvency is one of the most widely used and effective strategies to overcome the challenge of low aqueous solubility of many drugs and other compounds of interest. This technique involves the addition of a water-miscible organic solvent (the cosolvent) to the aqueous medium. The underlying principle is to modify the overall properties of the solvent medium by favoring solute–solvent interactions, reducing the interfacial tension between the hydrophobic solute and the aqueous medium, and/or reducing the energy required to create a cavity in the solvent to accommodate the solute, all of which result in an increase in the solubility of the poorly soluble compound [21,22].

In this context, the present study focuses on the binary co-solvent system formed by polyethylene glycol 200 (PEG 200) and water. PEG 200 is a low molecular weight polyethylene glycol with an average molar mass of about 190–210 g/mol, which is a viscous liquid at room temperature [23]. One of its most important properties is its complete miscibility with water in all proportions. These properties, together with its low toxicity, biocompatibility and wide use as a pharmaceutical excipient (solvent, solubilizer, humectant, vehicle in oral, topical and parenteral formulations), make it a cosolvent of great practical interest [24]. In addition, PEG 200 is considered a relatively “green” solvent due to its biodegradability and low volatility, which minimizes environmental and exposure risks [25].

Despite the pharmaceutical and environmental relevance of both TCC and PEG 200, and the utility of PEG 200/water mixtures as cosolvent systems, there is a notable lack of published experimental data on the equilibrium solubility of TCC specifically in these binary mixtures. While solubility data are available for TCC in other cosolvent systems, such as *N*-methyl-2-pyrrolidone/water [26] or ethanol/propylene glycol [17], and solubility studies exist for other drugs (such as naproxen [27], isoniazid [28], sildenafil [29] or fluphenazine [30]) and excipients (such as sucrose) in PEG 200/water mixtures, no research has been developed for the TCC–PEG 200–water system at different temperatures and cosolvent compositions. Therefore, the main objective of this research is to thermodynamically evaluate the solubility of Triclocarban (TCC) in polyethylene glycol 200 (PEG 200) + water cosolvent mixtures.

This study provides physicochemical information on the solubility of TCC, reporting fundamental experimental data that are currently unavailable, from which the thermodynamic analysis of the dissolution process is performed. From a practical point of view, this information is crucial for the rational design of pharmaceutical formulations, especially topical ones, containing TCC and using PEG 200. Finally, these data will allow a better understanding and prediction of the environmental behavior of TCC, including its fate, transport and bioavailability in aquatic systems.

## 2. Results and Discussion

### Experimental Solubility ($x_{3}$)

The experimental solubility data of TCC in cosolvent mixtures {PEG 200 (1) + W (2)} are presented in Table 1 and Figure 2.

Triclocarban (TCC) exhibits very limited solubility in aqueous systems. Although TCC is a predominantly hydrophobic molecule with low polarity, characterized by a Hildebrand solubility parameter of 21.97 MPa^1/2^ [26], it contains functional groups capable of specific interactions. These include the urea N-H groups, which can act as hydrogen bond donors (Lewis acids), and the carbonyl (C=O) oxygen, which can act as a hydrogen bond acceptor (Lewis base). However, a significant factor contributing to its low aqueous solubility is the high melting temperature and fusion enthalpy of TCC. These elevated thermal properties indicate strong intermolecular interactions within the crystalline solid state ($w_{22}$), which require significant energy input to overcome during dissolution. In addition, as a highly polar solvent with a high cohesive energy density, water can form ordered clusters [31]. This structured water network can hinder favorable interactions with the TCC molecule, ultimately reducing its solubility due to the “squeezing out” effect of water, also known as hydrophobic hydration [32,33].

Table 2 presents the fusion thermophysical properties of the commercial sample and bottom solid phases in equilibrium with the solution saturated in water, $w_{0.50}$ of PEG 200 and pure PEG 200. It is observed that the data reported in the present work are in agreement with the experimental data reported by other authors [26,34,35,36]. When comparing the DSC of the solid phases in equilibrium (water, PEG 200 $w_{0.50}$ and pure PEG 200) with the commercial solid phase (Figure 3), similar peaks are observed without drastic changes in the thermograms, indicating that polymorphic transformations and/or solvate formation did not occur due to changes in the cosolvent composition [37,38].

## 3. Solution Thermodynamic Functions of TCC in Cosolvent Mixtures PEG 200 + W

From the TCC solubility data in PEG 200 + W cosolvent mixtures, the thermodynamic solution functions were calculated using the Gibbs–van’t Hoff–Krug model according to [14,40,41]:(1)$$\Delta_{\mathrm{soln}}H{}^{\circ}=-R{\left(\frac{\partial \ln x_{3}}{\partial (T^{-1}-T_{\mathrm{hm}}^{-1})}\right)}_{p}=-R\cdot m$$(2)$$\Delta_{\mathrm{soln}}G{}^{\circ}=-RT_{\mathrm{hm}}\cdot \mathrm{intercept}$$(3)$$\Delta_{\mathrm{soln}}S{}^{\circ}=\left(\Delta_{\mathrm{soln}}H{}^{\circ}-\Delta_{\mathrm{soln}}G{}^{\circ}\right)T_{\mathrm{hm}}^{-1}$$(4)$$\zeta_{H}=|\Delta_{\mathrm{soln}}H{}^{\circ}\left|(\right|\Delta_{\mathrm{soln}}S{}^{\circ}\left|+\right|\Delta_{\mathrm{soln}}H{}^{\circ}{\left|\right)}^{-1}$$(5)$$\zeta_{TS}=1-\zeta_{H}$$
where $\Delta_{\mathrm{soln}}G{}^{\circ}$, $\Delta_{\mathrm{soln}}H{}^{\circ}$, $\Delta_{\mathrm{soln}}S{}^{\circ}$ and $T_{\mathrm{hm}}\Delta_{\mathrm{soln}}S{}^{\circ}$, are the Gibbs energy (in kJ·mol^−1^), enthalpy (in kJ·mol^−1^) and entropy (in kJ·mol^−1^·K^−1^) of the solution. $\zeta_{H}$ and $\zeta_{TS}$ describe the contribution of the energetic and organizational components to the value of the Gibbs energy of the solution. As for $T_{\mathrm{hm}}$, called harmonic mean temperature, it is the harmonic mean of the study temperatures [26].

From the van’t Hoff–Krug equation, *m* and *a* are calculated, from which $\Delta_{\mathrm{soln}}G{}^{\circ}$ and $\Delta_{\mathrm{soln}}H{}^{\circ}$ are calculated according to Equations (Equation 1) and (Equation 2) [34].(6)$$\ln x_{3}=m\cdot \left(T^{-1}-T_{\mathrm{hm}}^{-1}\right)+a$$

In Table 3, the values of the thermodynamic functions of the solution processes are presented: the Gibbs energy of the solution decreases from pure water to pure PEG 200, indicating a higher affinity of TCC with PEG 200. This behavior is similar to that reported in other investigations where the solubility is analyzed in aqueous cosolvent systems [26]; this behavior is due to the low solubility of TCC in mixtures rich in water and the subsequent increase in solubility of TCC when the cosolvent is added. When evaluating the enthalpy of solution, it is positive in all cases, indicating an endothermic process [42,43], which explains the increase in solubility with increasing temperature, favoring the breaking of TCC-TCC bonds due to the increase in molecular agitation; on the other hand, the decrease in the values of the enthalpy of solution from pure water to PEG 200 could mean that the energy of the solvent–solvent interactions decreases as the concentration of PEG 200 in the mixture increases, which in turn favors the formation of the cavity to accommodate the solute molecule (TCC). When evaluating the behavior of the solution entropy, negative values are observed between pure water and $w_{1}=0.20$, possibly due to the hydrophobic hydration effect of water in water-rich systems, causing a structuring of the water around the non-polar groups of the TCC; as the water concentration in the system decreases, the solute–solvent molecular interactions increase, improving the solubility of the TCC. When evaluating the contribution of enthalpy ($\zeta_{H}$) and entropy ($\zeta_{TS}$) to the Gibbs energy, we observe a process strictly governed by the enthalpy of solution, which is corroborated using Perlovich’s plot (Figure 4). Thus, values located in sectors I, IV, V and VIII indicate enthalpic conduction and values located in sectors II, III, VI and VII indicate entropic conduction, in accordance with

Sector I: $\Delta_{\mathrm{soln}}H{}^{\circ}\;>\;T\Delta_{\mathrm{soln}}S{}^{\circ}$;Sector II: $T\Delta_{\mathrm{soln}}S{}^{\circ}\;>\;\Delta_{\mathrm{soln}}H{}^{\circ}$;Sector III: $\Delta_{\mathrm{soln}}H{}^{\circ}\;<\;0;T\Delta_{\mathrm{soln}}S{}^{\circ}\;>\;0;|T\Delta_{\mathrm{soln}}S{}^{\circ}\left|\;>\;\right|\Delta_{\mathrm{soln}}H{}^{\circ}|$;Sector IV: $\Delta_{\mathrm{soln}}H{}^{\circ}\;<\;0;T\Delta_{\mathrm{soln}}S{}^{\circ}\;>\;0;|\Delta_{\mathrm{soln}}H{}^{\circ}|\;>\;|T\Delta_{\mathrm{soln}}S{}^{\circ}|$;Sector V: $\Delta_{\mathrm{soln}}H{}^{\circ}\;<\;0;T\Delta_{\mathrm{soln}}S{}^{\circ}\;<\;0;|\Delta_{\mathrm{soln}}H{}^{\circ}|\;>\;|T\Delta_{\mathrm{soln}}S{}^{\circ}|$;Sector VI: $\Delta_{\mathrm{soln}}H{}^{\circ}\;<\;0;T\Delta_{\mathrm{soln}}S{}^{\circ}\;<\;0;|T\Delta_{\mathrm{soln}}S{}^{\circ}\left|\;>\;\right|\Delta_{\mathrm{soln}}H{}^{\circ}|$;Sector VII: $\Delta_{\mathrm{soln}}H{}^{\circ}\;>\;0;T\Delta_{\mathrm{soln}}S{}^{\circ}\;<\;0;|T\Delta_{\mathrm{soln}}S{}^{\circ}\left|\;>\;\right|\Delta_{\mathrm{soln}}H{}^{\circ}|$;Sector VIII: $\Delta_{\mathrm{soln}}H{}^{\circ}\;>\;0;T\Delta_{\mathrm{soln}}S{}^{\circ}\;<\;0;|\Delta_{\mathrm{soln}}H{}^{\circ}|\;>\;|T\Delta_{\mathrm{soln}}S{}^{\circ}|$.

According to Figure 4, all points are located in sector I and sector VIII, which indicates that the solution process in all cases is driven by the enthalpy of the solution [44]

## 4. Thermodynamic Functions of Mixing of TCC in Cosolvent Mixtures PEG 200 + W

The dissolution process generally involves several sub-processes involving the hypothetical melting of the solute to subcooled liquid at the study temperature which is lower than the melting temperature, the restructuring of the solvent molecules to form the cavity to accommodate the solute and, finally, the accommodation of the solute molecule in subcooled liquid state in the cavity formed. This last process is the so-called mixing process whose energetics can be calculated as [45](7)$$\Delta_{\mathrm{soln}}H{}^{\circ}=\Delta_{\mathrm{mix}}H{}^{\circ}+\Delta_{\mathrm{fus}}H^{T_{\mathrm{hm}}}$$(8)$$\Delta_{\mathrm{soln}}H{}^{\circ}=\Delta_{\mathrm{mix}}H{}^{\circ}+\Delta_{\mathrm{fus}}S^{T_{\mathrm{hm}}}$$
where the thermodynamic quantities of the TCC melt (3) and its subsequent subcooling to $T_{\mathrm{hm}}$/K = 302.8 K are given by $\Delta_{\mathrm{fus}}H^{T_{\mathrm{hm}}}$ and $\Delta_{\mathrm{fus}}S^{T_{\mathrm{hm}}}$; these quantities are replaced by the enthalpy and entropy values of the ideal dissolution process, $\Delta_{\mathrm{soln}}{H{}^{\circ}}^{-\mathrm{id}}$ and $\Delta_{\mathrm{soln}}{S{}^{\circ}}^{-\mathrm{id}}$, which are taken from the literature [26].

In Table 4, the mixing thermodynamic functions of TCC in PEG 200 + W cosolvent mixtures are presented. Thus, from pure water to cosolvent mixture $w_{1}=0.55$, an enthalpic (positive values) and entropic (negative values) disadvantage to the mixing process is presented, i.e., the TCC molecules have a low affinity for the solvent molecules, especially for water, since as the concentration of PEG 200 in the cosolvent mixture increases, both the Gibbs energy and the entropy of mixing tend to decrease and the entropy of mixing tends to increase, indicating a higher affinity of TCC for the solvent. From $w_{1}=0.60$ up to pure PEG 200, the process is favored by the entropy of mixing (positive values) and disfavored by the enthalpy of mixing (positive values). Similar to the process in the first sector (from water up to $w_{1}=0.55$), the Gibbs energy and the enthalpy of mixing decrease, reflecting the affinity of TCC for PEG 200, reaching even negative values in mixtures rich in PEG 200 and pure PEG 200, where the solution process is favored by the mixing process.

When evaluating the contribution of enthalpy and entropy of mixing to the Gibbs energy of mixing using the Perlovich plot (Figure 5), from pure water up to $w_{1}=0.95$, the enthalpic contribution is dominant (sector I: $\Delta_{\mathrm{mix}}H{}^{\circ}\;>\;T\Delta_{\mathrm{mix}}S{}^{\circ}$ and sector VIII: $\Delta_{\mathrm{mix}}H{}^{\circ}\;>\;0;T\Delta_{\mathrm{mix}}S{}^{\circ}\;<\;0;|Delta_{\mathrm{mix}}H{}^{\circ}|\;>\;|T\Delta_{\mathrm{mix}}S{}^{\circ}|$ ). Although there is an entropic favoring from $w_{1}=60$, the enthalpy of mixing is higher. In other words, the energy required to break the solute–solute bonds and form the cavity to accommodate the solute is greater. In the cosolvent mixture with $w_{1}=0.95$ and pure PEG 200, the entropic contribution exceeds the enthalpy (Sector II: $T\Delta_{\mathrm{mix}}S{}^{\circ}\;>\;\Delta_{\mathrm{mix}}H{}^{\circ}$), which confirms the affinity of TCC for PEG 200.

## 5. Enthalpy–Entropy Compensation (EEC) Analysis

The solubility of TCC in PEG 200 + W cosolvent mixtures implies molecular interactions that generate an increase in the enthalpy of the solution, which is detrimental to the TCC dissolution process, but this enthalpic disfavor is clearly offset by the entropy generated, possibly by the non-covalent interactions between the TCC and the cosolvent system. Bustamante et al. indicate that EEC can be evaluated using a $\Delta_{\mathrm{mix}}H{}^{\circ}$ vs. $\Delta_{\mathrm{mix}}G{}^{\circ}$ plot, where positive slopes indicate enthalpic driving and negative slopes indicate entropic driving [46,47,48].

When analyzing the results presented in Figure 6, it is observed that the process in general is enthalpy-driven, since in all cases, the slope of the graph is positive.

## 6. Materials and Methods

### 6.1. Reagents

In this study, Triclocarban (Sigma-Aldrich, Burlington, MA, USA; compound **3**), Polyethylene glycol 200 (PEG 200) (Sigma-Aldrich, Burlington, MA, USA; the solvent component 1) and the double distilled water (component 2) with conductivity lower than 2 uS·cm^−1^, ethanol (Sigma-Aldrich, Burlington, MA, USA) were used. Table 5 summarizes the sources and purities of the compounds studied.

### 6.2. Experimental Procedure

The shake-flask method was used to determine the solubility of TCC in PEG 200 + W cosolvent mixtures [49,50,51].

#### 6.2.1. Preparation of Cosolvent Mixtures

The first step is to prepare the dissolution medium. By gravimetric preparation, 19 cosolvent mixtures (approximately 5 g) ranging from 0.05 to 0.95 in mass fraction were prepared in 10 mL amber bottles using an analytical balance with a sensitivity of ±0.0001 g (RADWAG AS 220.R2, Varsovia, Poland).

#### 6.2.2. Addition of Excess Solid Drug

A known amount of solid drug was added to the pure solvent (PEG 200/W) or cosolvent mixture contained in an amber glass vial with a tight-fitting screw cap.

An amount of solid drug was added to ensure the presence of an excess of undissolved solid phase once saturation equilibrium was reached. This excess is necessary to ensure that the solution is truly saturated. As a practical guide, it was verified that a quantity of solid was visibly present at the bottom of the vessel at the end of equilibration. An initial characterization of the solid state of the drug was performed by DSC because once equilibrium is reached, the solubility of the stable form at equilibrium is measured, which is not necessarily the initial form added.

#### 6.2.3. Sealing and Temperature Controlled Stirring

The vessel containing the solid–liquid mixture was hermetically sealed to prevent evaporation of the solvent, although PEG 200 and water are not highly volatile solvents.

The system was then subjected to continuous stirring to maximize contact between the surface of the solid and the solvent, thereby facilitating mass transfer and accelerating the approach to equilibrium.

After the initial agitation, the vials were placed in a thermostated recirculating water bath to maintain a constant temperature throughout the equilibration period. All vials were shaken regularly.

To determine the equilibrium time, the concentration of a sample was measured periodically until semi-constant results were obtained (variation no greater than 3%).

Note: Equilibration time was 36 h.

#### 6.2.4. Phase Separation

Once equilibrium was verified, the saturated liquid phase (supernatant) was separated from the excess undissolved solid prior to quantification. As this step is critical, it was performed with minimal disturbance of the established equilibrium.

At this point, an aliquot of the supernatant was taken with a syringe (after allowing the solid to settle briefly) and passed through a 0.45 µm membrane filter (Millipore Corp. Swinnex-13, Atlanta, GA, USA) attached to the syringe. To minimize drug loss by adsorption to the active sites of the filter, the first volume of filtrate (the first 0.1–0.5 mL) was discarded. In addition, the temperature was kept constant during filtration by preheating (or precooling, depending on the temperature of the experiment) the syringe and filter assembly to equilibrium temperature before sampling to avoid drug precipitation by cooling or additional dissolution by heating.

#### 6.2.5. Quantification

Finally, the concentration of TCC in the aliquot of saturated solution (filtrate or centrifuged supernatant) was quantitatively determined by UV-Vis spectrophotometry (UV/Vis EMC-11- UV spectrophotometer, Duisburgo, Germany), since TCC has a chromophore absorbing at 265 nm (wavelength of maximum absorbance). The Appendix A presents some data related to the calibration curve, including the linear equation, correlation coefficient, detection limit and quantification limit.

Due to the low solubility of TCC in aqueous systems, the saturated solution (S1) in these systems was determined by the standard addition technique, i.e., a solution of known concentration (S2) was prepared with an absorbance preferably at the midpoint of the calibration curve; then, 10.00 g of S1 (saturated solution) was mixed with 10.00 g of S2, which ensured the quantification of the resulting solution [26].

#### 6.2.6. Calorimetric Study

The enthalpy and melting temperature of four TCC samples were determined by differential scanning calorimetry (DSC 204 F1 Phoenix, Duisburgo, Germany). A mass of approximately 10.0 mg of each sample was deposited in aluminum crucibles and placed in the calorimeter under a nitrogen flow of 10 mL min^−1^. The heating cycle was developed from 323 to 523 K, with a heating ramp of 10 K min^−1^. This was adapted from [26].

## 7. Conclusions

The solubility of TCC in water-rich systems is low, possibly due to its hydrophobic nature, strong intermolecular interactions of the crystalline state of TCC, coupled with the high cohesive energy density of water and its tendency to form ordered clusters around the non-polar groups of TCC (hydrophobic hydration), which hinders the interactions between TCC and solvents. However, with increasing concentration of PEG 200, the solubility of TCC increases by several orders of magnitude, demonstrating a good affinity between TCC and PEG 200, which explains the decrease in Gibbs energy of the solution with increasing concentration of PEG 200 and the increase in entropy of mixing in PEG 200-rich mixtures. Although the enthalpy of solution is positive in all cases, there is a strong entropic compensation favoring the solution process, except in water-rich mixtures, where the structuring of water and the strong intermolecular interactions of TCC reduce the molecular mobility and therefore the solubility of TCC.

## Figures and Tables

**Figure 1 molecules-30-02631-f001:**
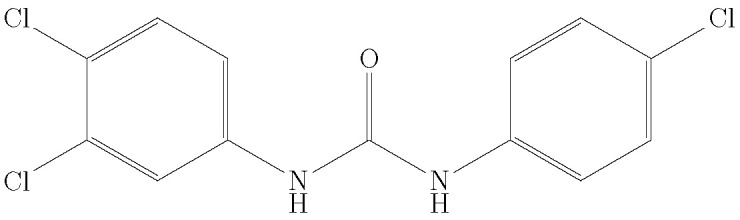
Molecular structure of Triclocarban.

**Figure 2 molecules-30-02631-f002:**
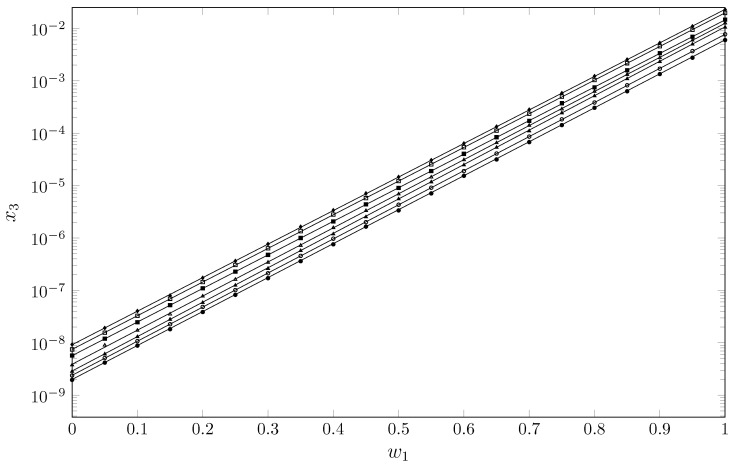
Mole fraction of TCC ($x_{3}$) depending on the mass fraction of PEG 200 in the {PEG 200 (1) + W (2)} mixtures free of TCC. •: 288.15 K; ∘: 293.15 K; ▴: 298.15 K; ▵: 303.15 K; ▪: 308.15 K; □: 313.15 K; ♦: 318.15 K.

**Figure 3 molecules-30-02631-f003:**
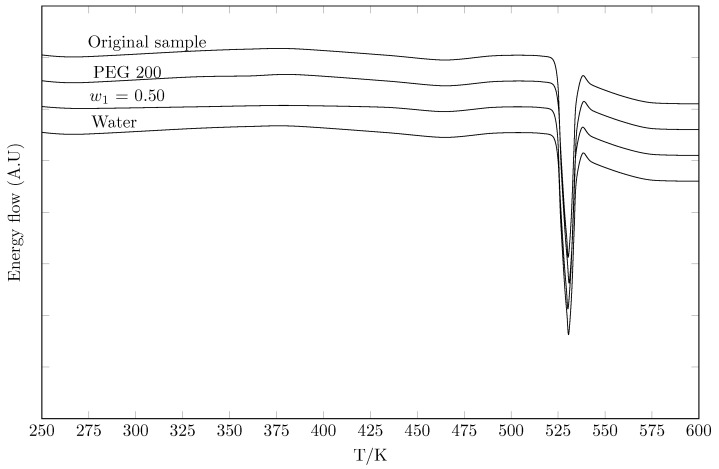
DSC thermograms of TCC.

**Figure 4 molecules-30-02631-f004:**
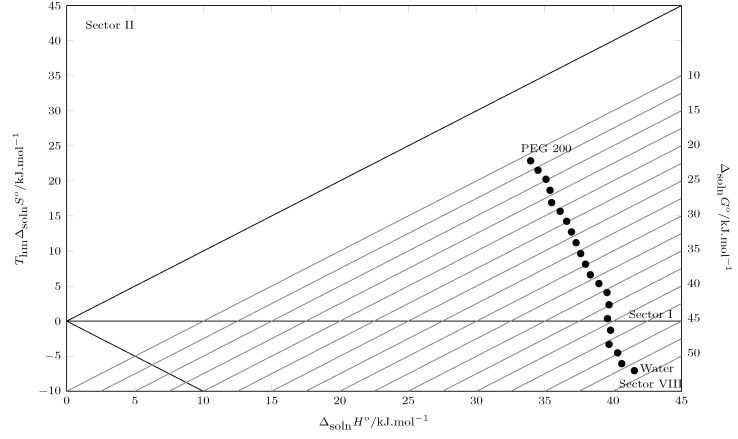
Relation between enthalpy ($\Delta_{\mathrm{soln}}H{}^{\circ}$) and entropy ($T\Delta_{\mathrm{soln}}S{}^{\circ}$) of the solution process of TCC (3) in PEG 200 (1) + W (2) cosolvent mixtures at 302.8 K. • = $T\Delta_{\mathrm{soln}}S{}^{\circ}$. (•: $T\Delta_{\mathrm{soln}}S{}^{\circ}$ values as a function of $\Delta_{\mathrm{soln}}H{}^{\circ}$). The isoenergetic curves for $\Delta_{\mathrm{soln}}G{}^{\circ}$ are represented by dotted lines.

**Figure 5 molecules-30-02631-f005:**
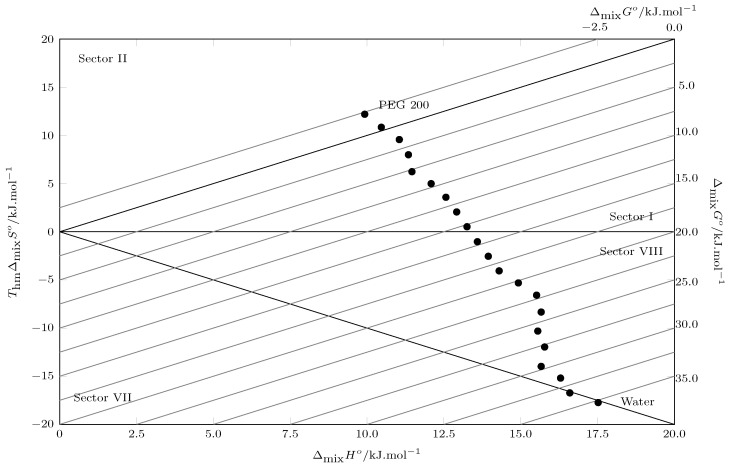
Relation between enthalpy ($\Delta_{\mathrm{mix}}H{}^{\circ}$) and entropy ($T\Delta_{\mathrm{mix}}S{}^{\circ}$) of the process mixing of TCC (3) in PEG 200 (1) + W (2) cosolvent mixtures at 302.8 K (•: $T\Delta_{\mathrm{mix}}H{}^{\circ}$ values as a function of $\Delta_{\mathrm{mix}}H{}^{\circ}$). The isoenergetic curves for $\Delta_{\mathrm{mix}}G{}^{\circ}$ are represented by dotted lines.

**Figure 6 molecules-30-02631-f006:**
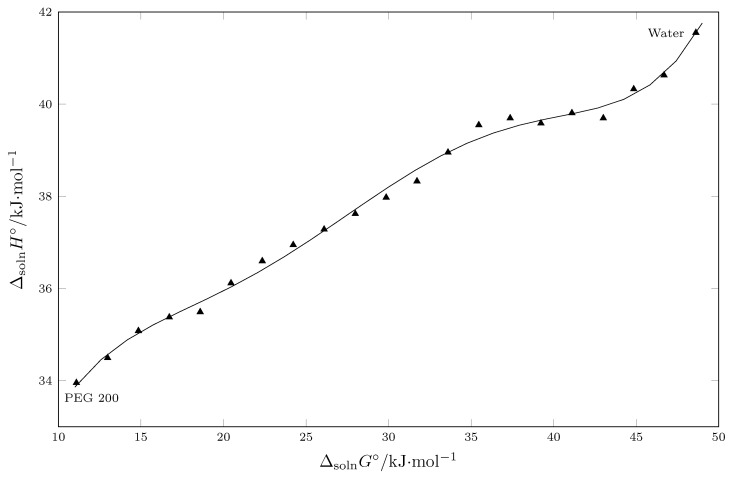
Enthalpy–entropy compensation plots for the solubility of TCC (3) in PEG 200 (1) + W (2) mixtures at $T_{\mathrm{hm}}$/K = 302.82 K. The points represent the mass fraction of PEG 200 (1) in the PEG 200 (1) + W (2) mixtures in the absence of TCC (3).

**Table 1 molecules-30-02631-t001:** Experimental solubility of TCC (3) in PEG 200 (1) + water (2) cosolvent mixtures expressed in mole fraction ($x_{3}$) at different temperatures. Average relative uncertainty in mole fraction solubility is $u_{r}\left(x_{3}\right)$ = 0.025; standard uncertainty in temperature is u(T) = 0.05 K. Experimental pressure *p*: 0.1 MPa. ($w_{1}$ is the mass fraction of PEG 200 (1) in the PEG 200 (1) + water (2) mixtures free of TCC, $u(w_{1})$ = 0.001).

*w* _1_	Temperatures						
	288.15 K	293.15 K	298.15 K	303.15 K	308.15 K	313.15 K	318.15 K
0.00	1.96$\cdot 10^{-9}$	2.38$\cdot 10^{-9}$	2.85$\cdot 10^{-9}$	3.78$\cdot 10^{-9}$	5.72$\cdot 10^{-9}$	7.48$\cdot 10^{-9}$	9.28$\cdot 10^{-9}$
0.05	4.19$\cdot 10^{-9}$	5.13$\cdot 10^{-9}$	6.11$\cdot 10^{-9}$	9.00$\cdot 10^{-9}$	1.20$\cdot 10^{-8}$	1.56$\cdot 10^{-8}$	1.92$\cdot 10^{-8}$
0.10	8.84$\cdot 10^{-9}$	1.08$\cdot 10^{-8}$	1.31$\cdot 10^{-8}$	1.73$\cdot 10^{-8}$	2.47$\cdot 10^{-8}$	3.28$\cdot 10^{-8}$	4.04$\cdot 10^{-8}$
0.15	1.82$\cdot 10^{-8}$	2.27$\cdot 10^{-8}$	2.80$\cdot 10^{-8}$	3.53$\cdot 10^{-8}$	5.23$\cdot 10^{-8}$	6.92$\cdot 10^{-8}$	7.98$\cdot 10^{-8}$
0.20	3.86$\cdot 10^{-8}$	4.84$\cdot 10^{-8}$	5.84$\cdot 10^{-8}$	7.77$\cdot 10^{-8}$	1.10$\cdot 10^{-7}$	1.44$\cdot 10^{-7}$	1.73$\cdot 10^{-7}$
0.25	8.23$\cdot 10^{-8}$	1.01$\cdot 10^{-7}$	1.25$\cdot 10^{-7}$	1.62$\cdot 10^{-7}$	2.29$\cdot 10^{-7}$	3.07$\cdot 10^{-7}$	3.64$\cdot 10^{-7}$
0.30	1.71$\cdot 10^{-7}$	2.13$\cdot 10^{-7}$	2.61$\cdot 10^{-7}$	3.44$\cdot 10^{-7}$	4.77$\cdot 10^{-7}$	6.37$\cdot 10^{-7}$	7.67$\cdot 10^{-7}$
0.35	3.61$\cdot 10^{-7}$	4.55$\cdot 10^{-7}$	5.72$\cdot 10^{-7}$	7.21$\cdot 10^{-7}$	1.00$\cdot 10^{-6}$	1.35$\cdot 10^{-6}$	1.64$\cdot 10^{-6}$
0.40	7.58$\cdot 10^{-7}$	9.71$\cdot 10^{-7}$	1.19$\cdot 10^{-6}$	1.57$\cdot 10^{-6}$	2.07$\cdot 10^{-6}$	2.80$\cdot 10^{-6}$	3.39$\cdot 10^{-6}$
0.45	1.65$\cdot 10^{-6}$	2.01$\cdot 10^{-6}$	2.52$\cdot 10^{-6}$	3.32$\cdot 10^{-6}$	4.38$\cdot 10^{-6}$	5.78$\cdot 10^{-6}$	7.10$\cdot 10^{-6}$
0.50	3.36$\cdot 10^{-6}$	4.32$\cdot 10^{-6}$	5.56$\cdot 10^{-6}$	6.88$\cdot 10^{-6}$	9.02$\cdot 10^{-6}$	1.22$\cdot 10^{-5}$	1.46$\cdot 10^{-5}$
0.55	7.10$\cdot 10^{-6}$	9.12$\cdot 10^{-6}$	1.17$\cdot 10^{-5}$	1.45$\cdot 10^{-5}$	1.90$\cdot 10^{-5}$	2.55$\cdot 10^{-5}$	3.05$\cdot 10^{-5}$
0.60	1.54$\cdot 10^{-5}$	1.90$\cdot 10^{-5}$	2.47$\cdot 10^{-5}$	3.10$\cdot 10^{-5}$	4.03$\cdot 10^{-5}$	5.41$\cdot 10^{-5}$	6.37$\cdot 10^{-5}$
0.65	3.16$\cdot 10^{-5}$	4.09$\cdot 10^{-5}$	5.34$\cdot 10^{-5}$	6.54$\cdot 10^{-5}$	8.37$\cdot 10^{-5}$	1.12$\cdot 10^{-4}$	1.34$\cdot 10^{-4}$
0.70	6.77$\cdot 10^{-5}$	8.62$\cdot 10^{-5}$	1.10$\cdot 10^{-4}$	1.41$\cdot 10^{-4}$	1.72$\cdot 10^{-4}$	2.35$\cdot 10^{-4}$	2.81$\cdot 10^{-4}$
0.75	1.43$\cdot 10^{-4}$	1.84$\cdot 10^{-4}$	2.44$\cdot 10^{-4}$	2.90$\cdot 10^{-4}$	3.73$\cdot 10^{-4}$	5.00$\cdot 10^{-4}$	5.80$\cdot 10^{-4}$
0.80	3.05$\cdot 10^{-4}$	3.85$\cdot 10^{-4}$	5.14$\cdot 10^{-4}$	6.26$\cdot 10^{-4}$	7.50$\cdot 10^{-4}$	1.03$\cdot 10^{-3}$	1.22$\cdot 10^{-3}$
0.85	6.31$\cdot 10^{-4}$	8.21$\cdot 10^{-4}$	1.09$\cdot 10^{-3}$	1.32$\cdot 10^{-3}$	1.59$\cdot 10^{-3}$	2.17$\cdot 10^{-3}$	2.54$\cdot 10^{-3}$
0.90	1.34$\cdot 10^{-3}$	1.70$\cdot 10^{-3}$	2.32$\cdot 10^{-3}$	2.76$\cdot 10^{-3}$	3.37$\cdot 10^{-3}$	4.56$\cdot 10^{-3}$	5.27$\cdot 10^{-3}$
0.95	2.76$\cdot 10^{-3}$	3.69$\cdot 10^{-3}$	4.98$\cdot 10^{-3}$	5.93$\cdot 10^{-3}$	6.92$\cdot 10^{-3}$	9.38$\cdot 10^{-3}$	1.09$\cdot 10^{-2}$
1.00	5.99$\cdot 10^{-3}$	7.72$\cdot 10^{-3}$	1.05$\cdot 10^{-2}$	1.27$\cdot 10^{-2}$	1.47$\cdot 10^{-2}$	1.98$\cdot 10^{-2}$	2.28$\cdot 10^{-2}$

**Table 2 molecules-30-02631-t002:** DSC thermograms of TCC as original sample and phases in equilibrium with saturated solutions.

Sample	Enthalpy of Fusion, $\Delta_{\mathrm{fus}}H/\mathrm{kJ}\cdot {\mathrm{mol}}^{-1}$	Melting Point, $T_{\mathrm{fus}}/K$	Ref.
Original sample	41.5±0.5	527.5.4±0.5	This work
	41.3±0.5	528.4±0.5	[26]
	41.94	528.2	[34]
		527.8	[35]
		525	[36]
		528.15–529.15	[39]
Water	41.1±0.5	528.5±0.5	This work
	41.9±0.5	527.5±0.5	[26]
PEG 200 ($w_{1}=0.50$)	42.2±0.5	527.9±0.5	This work
PEG 200	41.5±0.5	526.6±0.5	This work

**Table 3 molecules-30-02631-t003:** Apparent thermodynamic functions relative to solution processes of TCC (3) in PEG 200 (1) + W (2) mixtures at $T_{\mathrm{hm}}$/K = 302.8 as a function of the mass fraction of PEG 200 (1) ($w_{1}$) in the PEG 200 (1) + W (2) mixtures free of TCC (3).

$w_{1}$	$\Delta_{\mathrm{soln}}G{}^{\circ}$ /kJ·mol^−1^	$\Delta_{\mathrm{soln}}H{}^{\circ}$ /kJ·mol^−1^	$\Delta_{\mathrm{soln}}S{}^{\circ}$ /kJ·K^−1^·mol^−1^	$T_{\mathrm{hm}}\Delta_{\mathrm{soln}}S{}^{\circ}$ /kJ·mol^−1^	$\zeta_{H}$	$\zeta_{TS}$
0.00	48.61	41.55	−23.33	−7.06	0.85	0.15
0.05	46.69	40.63	−20.01	−6.06	0.87	0.13
0.10	44.85	40.33	−14.94	−4.52	0.90	0.10
0.15	43.01	39.70	−10.94	−3.31	0.92	0.08
0.20	41.11	39.81	−4.28	−0.30	0.97	0.03
0.25	39.23	39.59	1.18	0.36	0.99	0.01
0.30	37.37	39.70	7.69	2.33	0.94	0.06
0.35	35.46	39.55	13.48	4.08	0.91	0.09
0.40	33.60	38.95	17.68	5.35	0.88	0.12
0.45	31.72	38.33	21.82	6.61	0.85	0.15
0.50	29.85	37.98	26.82	8.12	0.82	0.18
0.55	27.98	37.62	31.82	9.64	0.80	0.20
0.60	26.09	37.28	36.95	11.19	0.77	0.23
0.65	24.22	36.95	42.01	12.72	0.74	0.26
0.70	22.35	36.60	47.05	14.25	0.72	0.28
0.75	20.45	36.12	51.72	15.66	0.70	0.30
0.80	18.59	35.49	55.82	16.90	0.68	0.32
0.85	16.72	35.38	61.62	18.66	0.65	0.35
0.90	14.85	35.08	66.82	20.23	0.63	0.37
0.95	12.98	34.49	71.05	21.51	0.62	0.38
1.00	11.09	33.95	75.51	22.87	0.60	0.40

**Table 4 molecules-30-02631-t004:** Apparent thermodynamic functions relative to mixing processes of TCC (3) in PEG 200 (1) + W (2) mixtures at $T_{\mathrm{hm}}$/K = 303.82 as a function of the mass fraction of PEG 200 (1) ($w_{1}$) in the PEG 200 (1) + W (2) mixtures free of TCC (3).

$w_{1}$	$\Delta_{\mathrm{mix}}G{}^{\circ}$ /kJ·mol^−1^	$\Delta_{\mathrm{mix}}H{}^{\circ}$ /kJ·mol^−1^	$\Delta_{\mathrm{mix}}S{}^{\circ}$ /kJ·mol^−1^·K^−1^	$T_{\mathrm{hm}}\Delta_{\mathrm{mix}}S{}^{\circ}$ /kJ·mol^−1^
0.00	35.25	17.52	−58.55	−17.73
0.05	33.32	16.60	−55.23	−16.73
0.10	31.49	16.30	−50.16	−15.19
0.15	29.64	15.67	−46.16	−13.98
0.20	27.74	15.78	−39.51	−11.96
0.25	25.87	15.55	−34.05	−10.31
0.30	24.00	15.67	−27.53	−8.34
0.35	22.10	15.52	−21.74	−6.58
0.40	20.24	14.92	−17.54	−5.31
0.45	18.36	14.30	−13.41	−4.06
0.50	16.49	13.95	−8.41	−2.55
0.55	14.62	13.59	−3.40	−1.03
0.60	12.73	13.25	1.72	0.52
0.65	10.86	12.92	6.79	2.06
0.70	8.99	12.57	11.82	3.58
0.75	7.09	12.09	16.49	4.99
0.80	5.23	11.46	20.59	6.24
0.85	3.35	11.35	26.39	7.99
0.90	1.48	11.05	31.60	9.57
0.95	−0.38	10.46	35.82	10.85
1.00	−2.27	9.92	40.28	12.20

**Table 5 molecules-30-02631-t005:** Source and purities of the compounds used in this research.

Chemical Name	CAS ^*a*^	Source	Purity in Mass Fraction	Analytic Technique ^*b*^
Triclocarban	101-20-2	Sigma-Aldrich	>0.990	HPLC
Polyethylene glycol 200	25322-68-3	Sigma-Aldrich	0.998	GC
Water	7732-18-5			
Ethanol	64-17-5	Sigma-Aldrich	0.998	GC

^*a*^ Chemical Abstracts Service Registry Number. ^*b*^ HPLC is high-performance liquid chromatography; GC is gas chromatography.

## Data Availability

Data are contained within the article or Appendix A.

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
