# Peer review of "Thermodynamic Study of the Solubility of Triclocarban in Polyethylene Glycol 200 + Water Cosolvent Mixtures at Different Temperatures"

_molecules, 2025, doi:10.3390/molecules30122631_

Round 1

Reviewer 1 Report

Comments and Suggestions for Authors

The study in question addresses a gap in the literature regarding TCC solubility in PEG 200 aqueous systems. The experimental work is extensive, covering a broad range of PEG 200 concentrations and temperatures. Also, a thorough thermodynamic analysis is performed using an established model. The manuscript is written in an informative manner, and its overall structure makes it is easy to follow. The experimental procedures follow standard protocols (shake-flask, UV-Vis, DSC) and are generally well-described, as is the used Gibbs-van’t Hoff-Krug thermodynamic model. The obtained results seem credible; however, there are some issues regarding the presentation of the results that need to be addressed.

  1. In line 117 “Table 4” should be “Table 3”.
  2. In line 136 “Figure ??” should be “Figure 4”.
  3. There is some confusion regarding the temperatures used in thermodynamic modeling. In Table 3 the temperature for thermodynamic functions relative to solution processes of TCC is listed as 297.6, in the description of Figure 4 it is 302.8, and later in the text it is 297.6 once again. In Table 4 the temperature for thermodynamic functions relative to mixing is listed as 303.82 and in the description of Figure 5 it is 302.8. Please clarify these inconsistencies or explain the rationale for the differences.
  4. In the description of Figure 5 the subscript “mez” should be “mix”
  5. The figure labeled as Figure 6 is incorrect, as it is a repeat of Figure 2.
  6. In section 6.2.5 please provide additional information about the calibration curve used for the spectrophotometric determination of TCC , including the linear equation, determination coefficient, and the limits of detection and quantification.
  7. There are some minor typos and errors in the text, e.g. “folloed” (line 8), “solublity” (line 15), “relative to of mixing processes” (Table 4). Please check the text carefully.

After addressing the above issues, I believe the manuscript will be suitable for publication in Molecules.

Reviewer 2 Report

Comments and Suggestions for Authors

In the manuscript entitled “Thermodynamic study of the solubility of Triclocarban in Polyethylene glycol 200 + water cosolvent mixtures at different temperatures” which authors are: Vanesa Puentes-Lozada, Diego Ivan Caviedes-Rubio, Cristian
Rincón-Guio, Nestor Enrique Cerquera, Rossember Edén Cardenas-Torres,
Claudia Patricia Ortiz, Fleming Martinez, Daniel Ricardo Delgado studied the solubility of TCC in cosolvent mixtures of PEG 200 + water at different temperatures. Experimental solubility data were determined using the shake-flask followed by UV quantification analysis at saturation methods, and thermodynamic functions of the solution processes were calculated using the Gibbs-van’t Hoff-Krug model.

Before the manuscript is accepted for publication, authors must make several modifications, such as:

1) There are some mistakes in English.

2) In Table 1, the authors present solubilities of the order of 10-9, and also present that the uncertainties of these solubilities are 0.025. Isn't there something wrong with that?

3) In line 136, Figure ??? appears.

4) Table 3 and Figure 4 are not mentioned in the manuscript.

5) In Figures 4 and 5, what do these black circles mean?

6) Although this work involves liquid and solid phases, the authors do not mention anything about pressure.

7) What is the uncertainty of the mass fractions represented by w?

8) What is the basis for dividing the diagram into sectors in the Perlovich plot? Does it have anything to do with solubilities?

9) In section 6.2.4, the authors said: Once equilibrium was verified ...... How was the solid-liquid equilibrium verified? The manuscript would be more complete if thermodynamic modeling of the ESL had been done, using some of the thermodynamic models suitable for these phases involved.

Comments on the Quality of English Language

Some mistakes in English should be corrected
